# Polyphenolic Boronates Inhibit Tumor Cell Proliferation: Potential Mitigators of Oxidants in the Tumor Microenvironment

**DOI:** 10.3390/cancers15041089

**Published:** 2023-02-08

**Authors:** Gang Cheng, Hakim Karoui, Micael Hardy, Balaraman Kalyanaraman

**Affiliations:** 1Department of Biophysics, Medical College of Wisconsin, Milwaukee, WI 53226, USA; 2Aix Marseille Univ, CNRS, ICR, 13009 Marseille, France

**Keywords:** polyphenols, mitochondria, tumor microenvironment, boronates, reactive and nitrogen species

## Abstract

**Simple Summary:**

Several plant-based compounds inhibit tumor cell growth. Selective modification of these compounds significantly increases their antitumor potency. The goal of this study was to develop less toxic, more potent natural compounds for use in cancer and immune therapies. To this end, we modified the structures of honokiol and magnolol, the two active components of magnolia extract. The modified compounds—honokiol boronate and mitochondria-targeted honokiol boronate—target the mitochondria of tumor cells and inhibit cell proliferation. These boronate derivatives also react with oxidants that are generated in tumor mitochondria and the tumor microenvironment. During this process, these boronate derivatives are also converted back to the original compounds with antitumor potencies. Thus, boronation of naturally occurring plant-derived compounds could make them more active in tumor cells as well as in the adjoining tumor microenvironment. These novel polyphenolic derivatives may enhance the scope of cancer immunotherapies.

**Abstract:**

Boronate-based compounds have been used in brain cancer therapy, either as prodrugs or in combination with other modalities. Boronates containing pro-luminescent and fluorescent probes have been used in mouse models of cancer. In this study, we synthesized and developed polyphenolic boronates and mitochondria-targeted polyphenolic phytochemicals (e.g., magnolol [MGN] and honokiol [HNK]) and tested their antiproliferative effects in brain cancer cells. Results show that mitochondria-targeted (Mito) polyphenolic boronates (Mito-MGN-B and Mito-HNK-B) were slightly more potent than Mito-MGN and Mito-HNK in inhibiting proliferation of the U87MG cell line. Similar proliferation results also were observed in other cancer cell lines, such as MiaPaCa-2, A549 and UACC-62. Independent in vitro experiments indicated that reactive nitrogen species (e.g., peroxynitrite) and reactive oxygen species (e.g., hydrogen peroxide) stoichiometrically react with polyphenolic boronates and Mito-polphenolic boronates, forming polyphenols and Mito-polyphenols as major products. Previous reports suggest that both Mito-MGN and Mito-HNK activate cytotoxic T cells and inhibit immunosuppressive immune cells. We propose that Mito-polyphenolic boronate-based prodrugs may be used to inhibit tumor proliferation and mitigate oxidant formation in the tumor microenvironment, thereby generating Mito-polyphenols in situ, as well as showing activity in the tumor microenvironment.

## 1. Introduction

Boronates are a novel class of molecules containing an electron-deficient boron atom. Borax and boronophenylalanine have been used in boron neutron capture therapy (BNCT) to treat glioma [1,2,3,4,5]. There is extensive literature indicating the antiproliferative and antitumor effects of several boron-containing compounds in various cancers [6,7,8,9,10,11,12]. In some studies, boronates are used as prodrugs, releasing the active antitumor drug in situ, which occurs following its reaction with hydrogen peroxide (H_2_O_2_) [6,7,8,9,10,11]. Boronate-based fluorophores and positron-emission tomography (PET) active compounds have been in cancer cells and cancer xenografts in the detection of oxidants [13,14,15,16].

From a chemical viewpoint, it has been known for decades that boronates react stoichiometrically yet slowly (k = 1 M^−1^s^−1^) with H_2_O_2_, forming the corresponding hydroxyl product [17]. Boron is electron-deficient and a weak Lewis acid. We and others have shown that biologically relevant proinflammatory oxidants such as peroxynitrite (ONOO^−^), hypochlorous acid (HOCl), and H_2_O_2_ undergo a nucleophilic addition reaction with phenylboronates and stoichiometrically form the corresponding phenol as a major product [13,18,19,20,21,22,23,24,25,26]. An interesting finding is that ONOO^−^ reacts nearly a million times faster with boronates than H_2_O_2_ [18,27,28]. Based on the rate constant determinations, boronates can compete very effectively with other biological reductants, such as glutathione, in their ability to scavenge ONOO^−^ [18].

There is increasing interest in understanding the role of oxidant formation and nitrosative stress in immunosuppressive tumor microenvironment (TME) [29,30,31,32,33,34,35,36,37,38,39]. Mitigating nitrosative stress has been reported to activate cytotoxic T cells involved in killing tumor cells [40]. Inhibition of cytokine nitration induced by reactive nitrogen species eliminated T cell exhaustion [41]. Reports also suggest that enhancing oxidant formation is critical for enhancing immunotherapy [35]. The paradoxical role of oxidative stress is discussed in relation to drug resistance in cancer [42].

In this study, we conjugated the boronate moiety to naturally occurring polyphenolic compounds and their mitochondria-targeted analogs (Figure 1). These boronate-based polyphenolics react with proinflammatory oxidants such as ONOO^−^, HOCl, and H_2_O_2_, and generate the original compounds in situ. These polyphenolics exert an anti-immune and antitumor function. In some cases, polyphenolic boronates themselves exhibit more potency than the parent polyphenolics. In addition, these compounds can stoichiometrically scavenge reactive oxygen and nitrogen species generated in tumors and in the TMEs from immune cells (e.g., neutrophils, macrophages). We surmise that a molecule incorporating the oxidative phosphorylation (OXPHOS)-inhibiting drug group attached to a boronate could decrease T cell exhaustion through scavenging of nitric oxide-derived oxidants released in the TME, thereby generating a polyphenolic mitochondria-targeted drug (MTD) with antitumor function. Here, we describe the results of polyphenolic boronates in glioblastoma, lung cancer, melanoma, and pancreatic cancer cell lines. The corresponding mitochondria-targeted polyphenols (mitochondria-targeted honokiol [Mito-HNK] and mitochondria-targeted magnolol [Mito-MGN]) have previously been tested in these cancer cells [43,44]. 

## 2. Materials and Methods

All chemicals and organic solvents were commercially available and were used as supplied. The reactions were monitored by thin layer chromatography (TLC) using silica gel Merck ^60^F254. Crude materials were purified by flash chromatography on Merck silica gel 60 (0.040–0.063 mm). ^31^P nuclear magnetic resonance (NMR), ^1^H NMR, and ^13^C NMR spectra were recorded with spectrometers at 400.13 MHz and 75.54 MHz, respectively. ^1^H NMR spectra were recorded at 400.13 MHz using a Bruker DPX AVANCE 400 spectrometer equipped with a quattro nucleus probe. ^1^H NMR and ^31^P NMR were taken in deuterated chloroform (CDCl_3_) using CDCl_3_ and tetramethylsilane as internal references, respectively. Chemical shifts (δ) are reported in ppm and *J* values in hertz. Synthetic procedures for preparing polyphenolic and Mito-polyphenolic boronates are shown in Figure 1, Figure 2, Figure 3, Figure 4 and Figure 5. NMR spectra are presented in Appendix A. 

### 2.1. Cell Culture

The U87MG (ATCC Cat# HTB-14, human glioblastoma cancer cells), A549 (ATCC Cat# CCL-185, human lung cancer cells), and MiaPaCa-2 (ATCC Cat# CRL-1420, human pancreatic cancer cells) cell lines were purchased from the American Tissue Culture Collection (Manassas, VA, USA). The UACC-62 melanoma cell line was purchased from AddexBio (San Diego, CA; Cat# C0020003). All cell lines were regularly authenticated. All cell lines were grown at 37 °C in 5% carbon dioxide. The U87MG, A549, and UACC-62 cells were maintained in Roswell Park Memorial Institute Medium 1640 medium (Thermo Fisher Scientific, Waltham, MA, USA; Cat# 11875) supplemented with 10% fetal bovine serum. MiaPaCa-2 cells were maintained in Dulbecco’s Modified Eagle Medium (Thermo Fisher Scientific, Waltham, MA, USA; Cat# 11965) and supplemented with 10% fetal bovine serum. All cells were stored in liquid nitrogen and used within 20 passages after thawing.

### 2.2. Cell Proliferation

The IncuCyte Live-Cell Analysis System was used to continuously monitor cell proliferation, as described in earlier publications [43,45,46]. 

### 2.3. Synthesis of HNK-B and HNK-B^2^

Honokiol boronate (HNK-B) and honokiol diboronate (HNK-B^2^) were prepared by reacting honokiol (HNK) in the presence of 2-(4-bromomethylphenyl)-4,4,5,5-tetramethyl-[1,3,2]-dioxaborolane and potassium carbonate in acetonitrile (MeCN). 

To a mixture of HNK (0.5 g, 1.92 mmol) and anhydrous potassium carbonate (0.34 g, 2.4 mmol) in MeCN (5 mL) was added 2-(4-bromomethylphenyl)-4,4,5,5-tetramethyl-[1,3,2]-dioxaborolane (0.57 g, 1.90 mmol). The mixture was stirred at reflux overnight. Then, ethyl acetate was added to the mixture as well as water (H2O) (20 mL). The organic layer was washed twice with H_2_O and dried over sodium sulfate (Na_2_SO_4_). The solvent was removed under reduced pressure. Purification by flash chromatography (pentane/diethyl ether [Et_2_O] 9/1 and 8/2) delivered the corresponding HNK-B as a mixture of two isomers (0.4 g, 43% yield) and HNK-B^2^ (0.1 g, 7%). High-resolution mass spectrometry (HRMS) calculated for HNK-B C_31_H_35_BO_4_ [M+Na]^+^ 505.2526 returned 505.2526. HRMS calculated for HNK-B^2^ C_44_H_52_B_2_O_6_ [M+Na]^+^ 721.3857 returned 721.3855.

#### 2.3.1. HNK-B

^1^H NMR (400.13 MHz, CDCl_3_) δ 7.83 (2H, 2d, *J* = 7.8, 7.8), 7.50–7.33 (3H, m), 7.25–7.12 (2H, m), 7.05–6.82 (3H, m), 6.09–5.94 (2H, m), 5.17 (2H,), 5.15–5.04 (5H, m), 3.48 (2H, 2d, *J* = 6.6, 6.3), 3.39–3.35 (2H, m), 1.36–1.37 (12H, 2s). ^13^C NMR (75 MHz, CDCl_3_) 156.1, 153.9, 153.2, 150.8, 140.5, 140.2, 137.8, 137.7, 136.5, 135.0, 134.9, 132.8, 132.1, 131.7, 131.2, 131.0, 130.7, 130.2, 130.1, 129.4, 129.0, 128.8, 127.9, 127.8, 126.3, 126.1, 124.7, 116.5, 115.9, 115.6, 115.5, 115.4, 113.5, 112.3, 83.9, 83.8, 70.6, 70.0, 39.4, 39.3, 35.2, 34.5, 29.7, 24.9.

#### 2.3.2. HNK-B^2^

^1^H NMR (400.13 MHz, CDCl_3_) δ 7.86 (2H, d, *J* = 7.8), 7.78 (2H, d, *J* = 7.8), 7.49 (2H, d, *J* = 7.8), 7.44 (1H, d, *J* = 1.8), 7.39–7.35 (3H, m), 7.15 (1H, d, *J* = 1.9), 7.06 (1H, dd, *J* = 8.3, 1.8), 6.93 (2H, dd, *J* = 8.3, 2.9), 6.07–5.92 (2H, m), 5.15 (2H, s), 5.13–5.04 (4H, m), 5.07 (2H, s), 3.49 (2H, d, *J* = 6.6), 3.37 (2H, d, *J* = 6.6), 1.37 (12H, s), 1.35 (12H, s). ^13^C NMR (75 MHz, CDCl_3_) δ 155.5, 153.9, 140.6, 140.5, 137.7, 136.9, 134.9, 134.8, 132.8, 131.3, 131.04, 131.02, 128.3, 128.3, 128.2, 127.8, 126.4, 126.1, 115.5, 113.4, 111.2, 83.8, 70.6, 69.9, 39.4, 34.6, 24.8.

### 2.4. Synthesis of Mito-HNK-B

Mitochondria-targeted honokiol boronate (Mito-HNK-B) was prepared by reacting Mito-HNK in the presence of 2-(4-bromomethylphenyl)-4,4,5,5-tetramethyl-[1,3,2]-dioxaborolane and potassium carbonate in MeCN. 

To a mixture of Mito-HNK (0.36 g, 0.48 mmol) and anhydrous potassium carbonate (0.1 g, 0.7 mmol) in MeCN (5 mL) was added 2-(4-bromomethylphenyl)-4,4,5,5-tetramethyl-[1,3,2]-dioxaborolane (0.24 g, 0.7 mmol). The mixture was stirred at reflux overnight. Then, dichloromethane (CH_2_Cl_2_) was added to the reaction mixture as well as H_2_O (20 mL). The organic layer was washed twice with H_2_O and dried over Na_2_SO_4_. The solvent was removed under reduced pressure. The residue was washed with Et_2_O. Purification by flash chromatography (from CH_2_Cl_2_ to CH_2_Cl_2_/ethanol [EtOH] 95/5) delivered the corresponding Mito-HNK-B as a mixture of two isomers as white solids (0.17 g, 37% yield). HRMS calculated for Mito-HNK-B C_59_H_69_BBrO_4_P [M]^+^ 883.5031 returned 883.5028. 

^31^P (400.13 MHz, CDCl_3_) δ 24.49, 24.46. ^1^H NMR (400.13 MHz, CDCl_3_) δ 7.90–7.64 (17H, m), 7.47–7.30 (4H, m), 7.16–7.00 (2H, m), 6.94–6.83 (2H, m), 6.07–5.90 (2H, m), 5.15–4.95 (6H, m), 3.88 (1H, t, *J* = 6.3), 3.90 (1H, t, *J* = 6.3), 3.85–3.73 (2H, m), 3.50–3.31 (4H, m), 1.82–1.68 (3H, m), 1.67–1.55 (5H,m), 1.35–1.34 (12H, 2s), 1.30–1.17 (8H, m). ^13^C NMR (75 MHz, CDCl_3_) δ 155.9, 155.3, 154.4, 153.9, 140.6, 137.8, 137.7, 137.1, 137.0, 134.94, 134.91, 134.89, 134.83, 133.7, 133.6, 132.7, 132.1, 131.3, 131.2, 131.1, 131.0, 130.8, 130.4, 130.3, 128.2, 128.0, 127.8, 127.7, 126.3, 126.1, 118.9, 118.0, 115.5, 115.4, 115.35, 115.32, 113.4, 112.7, 111.2, 110.7, 83.8, 83.7, 70.5, 69.9, 68.5, 67.9, 39.4, 34.6, 34.5, 30.4, 30.3, 29.7, 29.4, 29.3, 29.2, 29.1, 29.08, 26.1, 26.02, 24.9, 22.6 (d, *J* = 49.8), 22.5 (*J* = 4.4).

### 2.5. Synthesis of MGN-B

Magnolol boronate (MGN-B) was prepared by reacting magnolol (MGN) in the presence of 2-(4-bromomethylphenyl)-4,4,5,5-tetramethyl-[1,3,2]-dioxaborolane and potassium carbonate in MeCN.

To a mixture of MGN (0.5 g, 1.92 mmol) and anhydrous potassium carbonate (0.34 g, 2.4 mmol) in MeCN (5 mL) was added 2-(4-bromomethylphenyl)-4,4,5,5-tetramethyl-[1,3,2]-dioxaborolane (0.57 g, 1.90 mmol). The mixture was stirred at reflux overnight. Then, ethyl acetate was added to the mixture as well as H_2_O (20 mL). The organic layer was washed twice with H_2_O and dried over Na_2_SO_4_. The solvent was removed under reduced pressure. Purification by flash chromatography (pentane/Et_2_O 9/1 and 8/2) delivered the corresponding MGN-B (0.5 g, 55% yield). HRMS calculated for MGN-B C_31_H_35_BO_4_ [M+Na]^+^ 505.2526 returned 505.2528. 

^1^H NMR (400.13 MHz, CDCl_3_) δ 7.71 (2H, d, *J* = 7.8), 7.21 (2H, d, *J* = 7.9), 7.13–7.04 (3H, m), 7.02 (1H, d, *J* = 1.6), 6.95 (1H, d, *J* = 8.2), 6.91 (1H, d, *J* = 8.2), 6.14 (1H, s), 6.01–5.87 (2H, m), 5.08–5.01 (6H, m), 3.33 (4H, d, *J* = 3.3), 1.29 (12H, s). ^13^C NMR (75 MHz, CDCl_3_) δ 153.2, 151.9, 139.2, 137.9, 137.3, 135.0, 133.4, 132.5, 132.2, 131.1, 129.3, 129.1, 128.1, 126.4, 126.1, 117.3, 115.9, 115.5, 114.5, 83.8, 71.8, 39.4, 24.8.

### 2.6. Synthesis of Mito-MGN-B

Mitochondria-targeted magnolol boronate (Mito-MGN-B) was prepared by reacting Mito-MGN in the presence of 2-(4-bromomethylphenyl)-4,4,5,5-tetramethyl-[1,3,2]-dioxaborolane and potassium carbonate in MeCN. 

To a mixture of Mito-MGN (0.275 g, 0.37 mmol) and anhydrous potassium carbonate (0.076 g, 0.55 mmol) in MeCN (5 mL) was added 2-(4-bromomethylphenyl)-4,4,5,5-tetramethyl-[1,3,2]-dioxaborolane (0.14 g, 0.47 mmol). The mixture was stirred at reflux overnight. Then, CH_2_Cl_2_ was added to the reaction mixture as well as H_2_O (20 mL). The organic layer was washed twice with H_2_O and dried over Na_2_SO_4_. The solvent was removed under reduced pressure. The residue was washed with Et_2_O. Purification by flash chromatography (from CH_2_Cl_2_ to CH_2_Cl_2_/EtOH 95/05) delivered the corresponding Mito-MGN-B as a white solid (0.15 g, 42% yield). HRMS calculated for Mito-MGN-B (C_59_H_69_BBrO_4_P [M]^+^ 883.5031 returned 883.5032). 

^31^P (400.13 MHz, CDCl_3_) δ 24.32. ^1^H NMR (400.13 MHz, CDCl_3_) δ 7.85–7.64 (17H, m), 7.23–7.02 (6H, m), 6.87 (1H, d, *J* = 5.5), 6.85 (1H, d, *J* = 5.4), 6.01–5.89 (2H, m), 5.09–4.95 (6H, m), 3.81 (2H, t, *J* = 6.5), 3.75–3.66 (2H, m), 3.37–3.29 (4H, m), 1.55–1.43 (3H, m), 1.32–1.29 (12H, s), 1.25–1.21 (7H,m), 1.16–1.05 (6H, m). ^13^C NMR (75 MHz, CDCl_3_) δ 154.9, 154.4, 140.8, 137.85, 137.80, 134.95, 134.92, 134.62, 133.6, 133.5, 131.8, 131.79, 131.74, 131.4, 130.4, 130.3, 128.5, 128.2, 128.5, 127.9, 118.7, 117.9, 115.3, 112.8, 112.3, 83.7, 70.2, 68.5, 39.4, 30.4, 30.2, 29.3, 29.13, 29.10, 29.07, 28.9, 25.7, 24.9, 22.6 (d, *J* = 49.2), 22.5 (d, *J* = 4.4).

### 2.7. Synthesis of MAG-BET

Magnolol *ortho*-boronate (MAG-BET) was prepared by reacting MGN in the presence of 2-(2-bromomethylphenyl)-4,4,5,5-tetramethyl-[1,3,2]-dioxaborolane and potassium carbonate in MeCN. 

To a mixture of MGN (0.5 g, 1.92 mmol) and anhydrous potassium carbonate (0.34 g, 2.4 mmol) in MeCN (5 mL) was added 2-(2-bromomethylphenyl)-4,4,5,5-tetramethyl-[1,3,2]-dioxaborolane (0.57 g, 1.90 mmol). The mixture was stirred at reflux for 32 h. Then, ethyl acetate was added to the mixture as well as H_2_O (20 mL). The organic layer was washed twice with H_2_O and dried over Na_2_SO_4_. The solvent was removed under reduced pressure. Purification by flash chromatography (pentane/Et_2_O 9/1 and 8/2) delivered the corresponding MAG-BET (0.25 g, 27% yield). HRMS calculated for MAG-BET C_31_H_35_BO_4_ [M+Na]^+^ 505.2526 returned 505.2522. 

^1^H NMR (400.13 MHz, CDCl_3_) δ 7.85 (1H, d, *J* = 7.3), 7.41–7.28 (3H, m), 7.19–7.14 (2H, m), 7.11–7.04 (3H, m), 6.92 (1H, d, *J* = 8.1), 6.46 (1H, s), 6.06–5.94 (2H, m), 5.41 (2H, s), 5.15–5.03 (4H, m), 3.41–3.37 (4H, m), 1.31 (12H, s). ^13^C NMR (75 MHz, CDCl_3_) δ 153.5, 152.2, 142.4, 137.9, 137.5, 136.2, 133.7, 132.5, 132.0, 131.3, 131.2, 129.1, 128.9, 127.8, 127.6, 127.1, 126.5, 117.6, 115.7, 115.4, 113.8, 83.9, 71.1, 39.5, 39.4, 24.8. 

### 2.8. Calculated Values of the Octanol/Water Partition Coefficients

The calculated octanol/H_2_O partition coefficients (logP) of boronate analogs were assessed using a quantitative structure–activity relationship analysis and rational drug design as a measure of molecular hydrophobicity (Figure 1). This method also uses a consensus model built using the ChemAxon software (San Diego, CA, USA) [47,48].

### 2.9. LC/MS Analysis

High-performance liquid chromatography (HPLC) analysis was performed using an Agilent 1200 apparatus equipped with ultraviolet–visible spectroscopy absorption and a mass spectrometry detector (single quadrupole). 

In the studies on the reaction profile of ONOO^−^ oxidation of Mito-HNK-B and derivatives, 2 μL of sample was injected into the HPLC system equipped with a C18 column (Phenomenex, Kinetex Evo, 100 × 2.1 mm, 1.7 μm) equilibrated with 5% MeCN containing 0.1% (*v*/*v*) formic acid. The compounds were separated by a linear increase in MeCN phase concentration from 5% to 100% over 8 min and until 15 min at 100% MeCN using a flow rate of 0.21 mL/min. The peak areas detected by monitoring the absorption at 260 nm were used for the quantitation. 

### 2.10. Oxidation of Boronates Derivatives by ONOO^−^, H_2_O_2_, and HOCl 

The stock solutions of oxidants (HOCl and H_2_O_2_) were prepared freshly before each experiment; their concentrations were determined by spectrophotometry by reacting nitrite with H_2_O_2_, using the procedure described previously [15]. ONOO^−^ was prepared according to the published procedure. Typically, ONOO^−^ was synthesized in a reaction of 0.6 M nitrite with 0.7 M H_2_O_2_ at pH 13. Excess H_2_O_2_ was removed by passage through a column of MnO_2_ and the solution was frozen at −20 °C. The liquid over the frozen solid was collected and stored at −80 °C. Immediately prior to each experiment, the concentration of ONOO^−^ was determined spectrally at 302 nm (ε = 1.7 × 103 M^−1^cm^−1^) after dilution in 0.1 M sodium hydroxide to ~10 mM concentration [49]. 

Stock solutions of HNK-B, HNK-B^2^, and Mito-HNK-B were prepared in dimethyl sulfoxide at a 10 mM concentration and this solution was added directly to the buffer to obtain the desired concentration. HPLC analysis indicated that HNK-B, HNK-B^2^_,_ and Mito-HNK-B (pinacolate ester) undergo fast hydrolysis to the corresponding boronic acid formed upon dilution in the aqueous phosphate buffer. Thus, the boronic acid species were tested with the oxidants. When studying HNK-B and Mito-HNK-B oxidations by HOCl, stock solutions (10 mM) in MeCN were added to the phosphate buffer (100 mM, pH 7.4) to obtain a final concentration of the probe of 500 μM. Dimethyl sulfoxide solvent was avoided due to known rapid quenching of HOCl by dimethyl sulfoxide [50]. 

A study of the reactivity of HNK-B^2^ in the presence of sodium hypochlorite (NaOCl) was not possible because its solubility is too low in MeCN and water.

## 3. Results

### 3.1. Hydrophobicity of Polyphenolic Boronates 

Figure 1 lists the calculated partition coefficients for polyphenols (HNK, MGN, Mito-HNK, and Mito-MGN) and their boronate conjugates (HNK-B, HNK-B^2^, Mito_10_-HNK, Mito_10_-HNK-B, MGN-B, MGN-B^2^, Mito_10_-MGN, Mito_10_-MGN-B). Boronation increases the hydrophobicity of polyphenols. HNK-B^2^ is significantly more hydrophobic than HNK. 

### 3.2. Antiproliferative Effects of Polyphenolic Boronates in Glioblastoma, Melanoma, Pancreatic Cancer, and Lung Cancer Cell Lines

We compared the effects of polyphenols (MGN, HNK) and their boronate analogs (Mito-MGN-B and Mito-HNK-B) in the glioblastoma (U87MG), melanoma (UACC-62), pancreatic cancer (MiaPaCa-2), and lung cancer (A549) cell lines (Figure 2, Figure 3, Figure 4 and Figure 5). Figure 2 shows the dose-dependent effects of HNK and HNK-B in glioblastoma (U87MG), pancreatic cancer (MiaPaCa-2), and lung cancer (A549) cells. In these cell lines, the antiproliferative effects of HNK and HNK-B were nearly the same. Figure 3 shows the dose-dependent effects of Mito-HNK and Mito-HNK-B in these cell lines, and the antiproliferative effects of Mito-HNK and Mito-HNK-B were nearly the same. As shown in Figure 4, glioblastoma cells were treated with Mito-MGN/Mito-MGN-B and MGN/MGN-B. Boronation increased the potency of MGN-B as compared with MGN. Mito-MGN-B was also slightly more potent than Mito-MGN (Figure 4). In both cases, triphenylphosphonium (TPP^+^) conjugation (e.g., Mito-MGN and Mito-MGN-B) increased their antiproliferative potencies as compared with the parent compounds—MGN/MGN-B. Figure 5 shows the dose-dependent effects of MGN and MGN-B in decreasing the proliferation of melanoma (UACC-62) cells. 

### 3.3. Reaction between Polyphenolic Boronates and Oxidants 

Figure 6 and Figure 7 show the HPLC traces and mass spectral parameters for HNK/HNK-B, Mito-HNK/Mito-HNK-B, and corresponding intermediates and products formed from the reaction with oxidants. In the presence of ONOO^−^, both HNK-B and Mito-HNK-B were rapidly oxidized to HNK and Mito-HNK. In contrast, in the presence of H_2_O_2_, smaller conversion of HNK-B to HNK (60%) and Mito-HNK-B to Mito-HNK (37%) was observed. However, at higher concentrations of H_2_O_2_, this conversion was increased. These results are consistent with our original finding that ONOO^−^ reacts with boronates considerably faster than H_2_O_2_ [51]. Nitrated HNK and Mito-HNK were likely formed in minor quantities. HPLC results also show the formation of the hydroxy intermediate that triggers the self-immolative pathway (see Discussion, Figure 8).

## 4. Discussion

### 4.1. Oxidative Cleavage of Polyphenolic Boronates and TPP^+^-Conjugated Polyphenolic Boronates

The polyphenolic boronate compounds consist of an arylboronic ester head group and a quinone methide group linked via a carbamate group to the parent polyphenolic compound (e.g., HNK or Mito-HNK). In the presence of oxidants such as H_2_O_2_, HOCl, and ONOO^−^, boronates are oxidized through insertion of an oxygen atom into the carbon–boron bond, forming the corresponding phenol intermediate. The phenoxide ion eliminates the quinone methide, releasing the polyphenolic compound via a decomposition process known as self-immolation (Figure 8). As shown in our previous publications [13,25,26], ONOO^−^ reacts with boronates about one million times faster than H_2_O_2_ and one thousand times faster than HOCl. Phenols are the major product. Whereas the H_2_O_2_/boronate reaction is catalase-sensitive, the ONOO^−^/boronate reaction is catalase-insensitive. Peroxy-caged luciferin (PCL-1) is oxidized by H_2_O_2_, HOCl, and ONOO^−^ to luciferin as a major product through a self-immolation mechanism [15,20,27]. In an in vivo setting, PCL-1 has been used in luciferase-transfected mice xenografts to monitor oxidant formation through measurement of bioluminescence and in PET using a boronate-based positron emitting probe [16,20,21,22].

### 4.2. Antitumor Activity of MTDs: Activation of Immune Cells 

We showed that mitochondria-targeted atovaquone (Mito-ATO) targets both granulocytic-myeloid-derived suppressor cells (G-MDSCs) and regulatory T cells (T_regs_) in the TME, resulting in significant decreases of both G-MDSCs and T_regs_ as determined by flow cytometry analysis [45]. Intratumoral injection of Mito-ATO into primary tumors in a spontaneous tumor model triggered potent T cell immune responses locally and in distant tumor sites. Single-cell RNA sequencing revealed that Mito-ATO inhibits the expression of genes for OXPHOS and glycolysis in G-MDSCs and T_regs_ and facilitates the infiltration of CD4^+^ T cells in the tumor microenvironment [52]. Other studies also revealed the immuno-modulatory and tumor-preventing effects of MTDs [53,54]. 

### 4.3. Oxidants in the Tumor Microenvironment: Potential Inhibition by Polyphenolic Boronates 

The TME changes dynamically. The TME consists of tumor cells, cancer-associated macrophages, lymphocytes, neutrophils, cancer-associated fibroblasts, endothelial cells, and vascular pericytes. The extracellular matrix in the TME consist of protein and polysaccharides. The dynamic interaction or the lack thereof between the tumor cells and the many cells and components in the TME regulates metabolism. MDSCs generate reactive oxygen and nitrogen species in the TME via activation of nicotinamide adenine dinucleotide phosphate oxidase (NOX) and inducible nitric oxide synthase (iNOS) [55].

We have previously reported the anti-inflammatory effects of MTDs in Mito-Park mice [56,57]. MTDs such as Mito-apocynin decreased the expression of iNOS and NOX2, resulting in decreased oxidative and nitrative damage in neuronal cells. In another study, administration of an MTD (i.e., mitochondria-targeted carboxy-proxyl [Mito-CP]) inhibited cisplatin-induced renal toxicity by inhibiting pro-inflammatory mediators (iNOS in macrophages) in the kidney [58]. Mitochondria-targeted compounds inhibit oxidants generated from NOX2 [56,59]. Inhibition of NOX4 (which presumably generates only H_2_O_2_) potentiates cancer immunotherapy [60,61]. Recent clinical trials using iNOS inhibitors in breast cancer patients who were resistant to other forms of cancer therapies showed tumor shrinkage and enhanced patient survival [62]. Inhibition of nitric oxide generated in the TME from tumor-associated macrophages decreased inflammatory mediators in the TME and enhanced killing of cancer cells by T cells.

In the present study, the reaction between polyphenolic boronates and reactive oxygen/nitrogen species released the parent polyphenolic compounds in situ. When MTD is released, it may inhibit iNOS and nitric oxide generation in the TME, thus decreasing inflammation. Figure 9 shows the antioxidant potentials of Mito-HNK-B/Mito-MGN-B and HNK-B/MGN-B in the TME, especially the direct and nearly stoichiometric scavenging of ONOO^−^. The compound 3-(aminocarbonyl) furoxan-4-yl) methyl salicylate (AT38) was used to enhance the efficacy of immunotherapy in pancreatic cancer [40]. AT38 inhibited the expression of arginase-1 and NOS-2 (or iNOS) in myeloid cells in the TME [41]. AT38 mitigated chemokine nitration, promoting intra-tumoral infiltration of T cells. It is not clear if AT38 had any effect on NOX2 expression or on formation of superoxides and H_2_O_2_. Boronate-based drugs are likely to be more potent scavengers of ONOO^−^ (and HOCl), even in the presence of cellular reductants such as glutathione. Detection of the minor nitrated or chlorinated product from polyphenolic boronates can serve as a very reliable and diagnostic marker product of ONOO^−^ or HOCl generated in the TME. Preliminary results obtained from our laboratory indicate that Mito-HNK inhibits oxidant formation from NOX2 (Zielonka J and Kalyanaraman B, unpublished data). At present, the inhibitory effects of Mito-HNK or Mito-MGN or their boronate analogs on iNOS and NOX2 expression in cells are not known.

### 4.4. Mitochondria-Targeted Boronates as Radiosensitizers

Boronated agents (e.g., boronophenylalanine) are used to enhance the therapeutic ratio in neutron treatment of cancers. This BNCT modality involves selective delivery of boron-10 to tumors followed by irradiation with neutrons. During this process, lithium-7 and an alpha particle that are generated destroy the tumor tissues [1,2,3,4,5]. BNCT has been used clinically to treat cancers, including glioblastoma multiforme, head and neck cancers, lung cancer, and breast cancer. A critical requirement for enhanced therapeutic success is selective localization of boron-containing compounds in tumor tissues. Mitochondria-targeted boronates (Mito-HNK-B and Mito-MGN-B) that are selectively sequestered in tumor mitochondria could be more effective in capturing neutrons in mitochondria of tumors and enhance tumor killing. 

Inhibition of mitochondrial respiration could alleviate tumor and TME hypoxia and act as radiation sensitizers. Mitochondria-targeted metformin and X-radiation sensitized the killing of pancreatic cancer cells [46,63]. Clinical trials are ongoing with atovaquone and radiation therapy [64]. Tumor hypoxia inhibits the efficacy of radiation and immunotherapies in cancer. Targeting mitochondrial metabolism in non-small cell lung cancer patients was reported to modify both the tumor and TME through a decrease in hypoxia and hypoxic gene expression [65,66]. Mitochondria-targeted drugs decrease tumor hypoxia (or increase tumor oxygenation) by inhibiting mitochondrial respiration in tumors. Blood–brain-barrier-permeable aromatic boronates were effective in treating brain tumors using neutron capture therapy [1,6]. Although it is not known whether mitochondria-targeted polyphenolic boronates (Mito-MGN-B and Mito-HNK-B) are blood–brain-barrier permeable, Mito-HNK inhibited metastasis of lung tumors to the brain in mice xenografts and showed blood–brain-barrier permeability [43]. 

### 4.5. Mitochondria Metabolism and Racial Disparity in Cancer

Recent publications suggest that race and ethnicity are factors that affect mitochondrial metabolism in cancer cells, and that mitochondrial metabolism in white cancer patients is considerably different than in patients from other racial and ethnic groups [67,68,69,70]. Studies have revealed distinct differences in the TME in Black cancer patients. In Black patients with bladder cancer, mitochondrial metabolism is considerably higher [71]. Although more studies are needed to fully substantiate this trend, the differential metabolism provides a compelling rationale for testing mitochondria-targeted drugs in Black patients with bladder cancer.

The existence of racial and ethnic disparities has been identified in the breast cancer microenvironment [72,73,74]. The TME of Black cancer patients with breast cancer exhibits increased level pro-tumorigenic factors (e.g., macrophages, T_regs_, exhausted T cells) compared with white counterparts [67,68,72,75]. Upregulation of OXPHOS genes was detected in tumor samples isolated from Black cancer patients [76]. Tumors obtained from Black cancer patients have more mitochondria and PGC-1α (proliferation-activated receptor gamma coactivator 1-alpha) [69]. Identification of biomarkers (e.g., hypoxia-induced genes) in these patients may provide additional insights [65]. Black cancer patients were reported to exhibit a higher level of pro-inflammatory cytokines [76]. Clinical trials targeting mitochondrial metabolism in cancer should include Black patients and patients from diverse racial and ethnic backgrounds. Presently, metformin is used as the sole drug of choice to test the mechanistic role of mitochondria in cancer disparity studies [67,77]. Metformin is one of the most prescribed drugs for treating diabetes. Despite increased safety, its bioavailability is poor. Clinical trials revealed that Black cancer patients respond better to mitochondrial OXPHOS inhibitors (e.g., metformin) than white cancer patients [67,77]. The present study opens the possibility of testing a new class of mitochondria-targeted drugs that target both the tumor mitochondria and TME to treat Black cancer patients. 

## 5. Conclusions

The development of drugs that target both the tumor mitochondria and the TME would be a value-adding therapeutic advancement [78,79,80,81,82]. Higher expression of OXPHOS in triple-negative breast cancer patients who received neoadjuvant chemotherapy was associated with worse treatment outcome [83]. In this study, we developed a novel class of mitochondria-targeted polyphenolic boronates that inhibit tumor cell proliferation and scavenge oxidants such as H_2_O_2_, HOCl, and ONOO^−^ that are generated in the immunosuppressive TME. During this reaction, immunoactive parent polyphenols are regenerated. Specific polyphenols include HNK and MGN, which are active components of magnolia plant extract. Borono-L-phenylalanine was used to treat brain cancer in humans, so it is likely that polyphenolic boronates and the mitochondria-targeted boronate analogs are nontoxic and have high clinical and translational potential. We hope that the redox-based mitochondria-targeted polyphenolic boronates will be tested in the appropriate mice models (e.g., KPC mice) to assess their ability to inhibit reactive oxygen species/reactive nitrogen species-mediated nitration in the TME. 

## Data Availability

Not applicable.

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
