# Peer review of "Polyphenolic Boronates Inhibit Tumor Cell Proliferation: Potential Mitigators of Oxidants in the Tumor Microenvironment"

_cancers, 2023, doi:10.3390/cancers15041089_

Round 1

Reviewer 1 Report

This manuscript presents the characterization of synthesized polyphenolic boronates and their effects on various cell tumoral lines and the calculation of the IC50.

 Minor comments

1.     Because this manuscript is based more on the synthesis and identification of polyphenolic and mito-polyphenolic boronates used in their final characterization, together with the internal references, please, add some details about the validation methods (some performance indicators in the laboratory), which are necessary to improve the quality of the methods and the manuscript.

 2.      All cell lines presented in the material and methods section are used in the experiments (human brain cancer cells, human lung cancer cells, melanoma cell lines)? In the results section, you presented glioblastoma, pancreatic cancer, lung cancer cell lines, etc. Please revise the material and methods chapter to have in the entire manuscript the same cell lines.

 3.      About the HPLC/UV-Vis/MS analyses, because you didn’t use the standardized methods and used a synthesized internal reference, please add some performance indicators by method validation in the laboratory to sustain your method development “in the house”.

 4.      In figure 6, please correct part A to see the writing of the HPLC peaks and retention time.

 5.      Please revise the discussion because it is based on the activation of immune cells (T cells) by flow cytometry or inflammatory effects (your previous studies), and in this manuscript, you studied the IC50 after your applied derived boronates on reference cell lines. To improve the quality of the manuscript, I suggest adding some flow cytometry experiments about the total oxidative stress or mitochondrial stains to observe the direct effect of these antioxidants on the mitochondria, not only at theoretical levels. Suppose it isn’t possible to add these new flow cytometry experiments to confirm your theoretical approach. In that case, I suggest adding more references about the effects of the polyphenols, such as antioxidants, in cancer cell line proliferation.

 6.      Please revise the conclusions to correlate with your observed results. 

Author Response

We thank the reviewer for the positive remarks and constructive comments. We have revised the manuscript, taking into consideration the critiques as best as we can. Our point-by-point response follows:  

1. Because this manuscript is based more on the synthesis and identification of polyphenolic and mito-polyphenolic boronates used in their final characterization, together with the internal references, please, add some details about the validation methods (some performance indicators in the laboratory), which are necessary to improve the quality of the methods and the manuscript.

Response: In addition to the NMR and mass spectral characterization of mito-polyphenolic boronates, we used the reaction with hydrogen peroxide as a functional indicator. As shown, nearly stoichiometric formation of mito-HNK was observed in the reaction between mito-HNK-boronate and H2O2.

2. All cell lines presented in the material and methods section are used in the experiments (human brain cancer cells, human lung cancer cells, melanoma cell lines)? In the results section, you presented glioblastoma, pancreatic cancer, lung cancer cell lines, etc. Please revise the material and methods chapter to have in the entire manuscript the same cell lines.

Response: As suggested, we have revised the methods and results sections and reconciled all cell lines in the manuscript. We have also added “melanoma cell lines” in the results section.

3. About the HPLC/UV-Vis/MS analyses, because you didn’t use the standardized methods and used a synthesized internal reference, please add some performance indicators by method validation in the laboratory to sustain your method development “in the house”.

Response: As discussed in response to #1, the performance indicator is the reaction between the mito-polyphenolic boronates and H2O2 forming the parent polyphenols.

4. In figure 6, please correct part A to see the writing of the HPLC peaks and retention time.

Response: Done.

5. Please revise the discussion because it is based on the activation of immune cells (T cells) by flow cytometry or inflammatory effects (your previous studies), and in this manuscript, you studied the IC50 after your applied derived boronates on reference cell lines. To improve the quality of the manuscript, I suggest adding some flow cytometry experiments about the total oxidative stress or mitochondrial stains to observe the direct effect of these antioxidants on the mitochondria, not only at theoretical levels. Suppose it isn’t possible to add these new flow cytometry experiments to confirm your theoretical approach. In that case, I suggest adding more references about the effects of the polyphenols, such as antioxidants, in cancer cell line proliferation.

Response: As suggested, we have included additional references on the antiproliferative effects pf polyphenols in cancer cells.

6. Please revise the conclusions to correlate with your observed results.

Response: As suggested, we revised the conclusions to highlight the results obtained in the present work.

Reviewer 2 Report

Comments to the authors:

The article “Polyphenolic boronates inhibit tumor cell proliferation: Potential mitigators of oxidants in the tumor microenvironment” (Manuscript ID: cancers-2145103) focuses on the synthesis and analysis of plant-based compounds in the inhibition of cancer cells proliferation and oxidant formation in the tumor microenvironment as well as the possible chemical modifications in order to enhance their antitumor activity. The aim of this paper was to observe the effects of Honokiol boronate and Magnol boronate (polyphenolic boronates) and mitochondria-targeted polyphenolic phytochemicals, on different cancer cell lines. This is an interesting and modern topic since the importance of discovery new and natural antitumoral molecules. The submitted work is nicely written and well presented. To further improve your manuscript, some suggestions are listed below:

Point 1: The authors should include and specify in the abstract on what cell lines the experiments were made.

Point 2: Why the authors decided to test these compounds on Brain, Lung, and Pancreatic Cancer Cell Lines? Please explain this in the introduction. In that section, you only mentioned the applications of boronates compounds in glioma therapy, without mentioning the possible applications for lung and pancreatic cancers and melanoma too.

Point 3: The first scheme where the Synthesis of HNK-B and HNK-B2 is reported, results to be not so clear. The authors should better indicate the name of the compounds underneath each structure.

Point 4: Figures 2, 3 and 4 result to be blurred and pixelated. Please, check and improve the quality of these images. Remember that high quality images should never appear pixelated when zooming in it.

Point 5: In section 3.2, the authors should add “melanoma cell lines” in the title of the section.
Point 6: The antiproliferative effects of these modified compounds results to be very intriguing, but further investigations are required to better understand if we are facing cellular death or just a proliferation arrest. Firstly, I suggest the authors to provide the manuscript of a BrdU assay for all the cell lines used in order to investigate the cell proliferation. Secondly the authors should also analyze cell death by a flow cytometry apoptosis assay for all cell lines used.

Point 7: The authors should insert the abbreviations list to the manuscript and always specify in the acronym in the text in order to make it easier.

Point 8: The authors should explain better in the introduction why the mitochondria targeting is so important. Mitochondrial functioning results to be impaired in glioblastoma (DOI: 10.3390/cancers14246044) and melanoma too (DOI: 10.3390/biom10101395). Starting from the reported articles, please add some more information.

Author Response

We thank the reviewer for the positive remarks and constructive comments. We have revised the manuscript, taking into consideration the critiques as best as we can. Our point-by-point response follows:  

1. The authors should include and specify in the abstract on what cell lines the experiments were made.

Response: Done.

2. Why the authors decided to test these compounds on Brain, Lung, and Pancreatic Cancer Cell Lines? Please explain this in the introduction. In that section, you only mentioned the applications of boronates compounds in glioma therapy, without mentioning the possible applications for lung and pancreatic cancers and melanoma too.

Response: We have included in the Introduction the reasons for testing the polyphenolic boronates in brain, lung, and pancreatic cancer cells.

We included a sentence mentioning the potential applications to other cancers as suggested.

3. The first scheme where the Synthesis of HNK-B and HNK-B2 is reported, results to be not so clear. The authors should better indicate the name of the compounds underneath each structure.

Response: We have included the compound name beneath the chemical structure in scheme 1.

4. Figures 2, 3 and 4 result to be blurred and pixelated. Please, check and improve the quality of these images. Remember that high quality images should never appear pixelated when zooming in it.

Response: We have replaced Figures 2, 3, and 4 (as well as Figure 5) to improve the image quality.

5. In section 3.2, the authors should add “melanoma cell lines” in the title of the section.

Response: We have made this change as suggested.

6. The antiproliferative effects of these modified compounds results to be very intriguing, but further investigations are required to better understand if we are facing cellular death or just a proliferation arrest. Firstly, I suggest the authors to provide the manuscript of a BrdU assay for all the cell lines used in order to investigate the cell proliferation. Secondly the authors should also analyze cell death by a flow cytometry apoptosis assay for all cell lines used.

Response: Although there are some limitations as indicated by the reviewer, the advantages of the IncuCyte Live-Cell Analysis System are that  it is probe-free and noninvasive, and enables continuous monitoring of cell confluence over several days. As shown in one of our publications (Cheng G et al. Canc Treat Res Commun_2020_25 10021; AbuEid M et al. iScience. 2021 25;24(6):102653), the antiproliferative effects are related to cell cycle arrest. Using the SYTOX Green assay, we showed that mitochondria-targeted drugs alone did not cause cell death at the low micromolar concentrations.    

7. The authors should insert the abbreviations list to the manuscript and always specify in the acronym in the text in order to make it easier.

Response: We have included abbreviations in the same order throughout; additionally, we have included a list of abbreviations as the reviewer suggested.

8. The authors should explain better in the introduction why the mitochondria targeting is so important. Mitochondrial functioning results to be impaired in glioblastoma (DOI: 10.3390/cancers14246044) and melanoma too (DOI: 10.3390/biom10101395). Starting from the reported articles, please add some more information.

Response: We revised the Introduction to indicate why mitochondrial targeting is important and also have discussed the results from the papers suggested by this reviewer. 

Reviewer 3 Report

Manuscript entitled ‘Polyphenolic boronates inhibit tumor cell proliferation: Potential mitigators of oxidants in the tumor microenvironment’ written by Gang Cheng et al. have modified the structures of honokiol and magnolol, the two active components of magnolia extract and shown that the modified compounds inhibit cancer cell proliferation in different cell lines. Moreover, authors have shown that the boronate derivatives also react with oxidants that are generated in tumor mitochondria and the tumor microenvironment by HPLC. 

However, there are several major concerns regarding this manuscript.  

1. The authors showed that modified compounds such as Mito-MGN inhibited cancer cell proliferation in a variety of cell lines, however, another publication by the authors reported similar results but only tested on different cell lines. (Title: Mitochondria-targeted magnolol inhibits OXPHOS, proliferation, and tumor growth via modulation of energetics and autophagy in melanoma cells). 

2. The authors have shown that the boronate derivatives also react with oxidants (H2O2, NaOCl, and ONOO−) that are generated in tumor mitochondria and the tumor microenvironment by HPLC, but have not taken any further investigations to study the consequence of this reaction in cancer cells, thus no results to support any claim in discussion e.g. these novel polyphenolic derivatives may enhance the scope of cancer immunotherapies’.

3. The main results of this manuscript focus on the chemical reactions that form borate derivatives, which are not really cancer-related findings.

All things considered, I'm afraid this manuscript is not ready for publication in Cancers.

Author Response

We thank the reviewer for the constructive comments. We have addressed the critiques as best as we can. Our point-by-point response follows:  

1. The authors showed that modified compounds such as Mito-MGN inhibited cancer cell proliferation in a variety of cell lines, however, another publication by the authors reported similar results but only tested on different cell lines. (Title: Mitochondria-targeted magnolol inhibits OXPHOS, proliferation, and tumor growth via modulation of energetics and autophagy in melanoma cells).

Response: The reviewer is correct in that Mito-MGN was shown to inhibit OXPHOS and tumor proliferation in pancreatic cancer cells. However, we did not use Mito-MGN boronate, a totally new redox active polyphenolic derivatives targeting mitochondria. We also showed in this study the results from Mito-honokiol boronates and honokiol boronates in various tumors. Testing in numerous cancer cell lines in addition to melanoma cells is important as some melanoma cells are more prone to inhibition by OXPHOS inhibitors. Thus, we have used totally new redox-active boronate derivatives of both mitochondria-targeted and untargeted polyphenols.  

2. The authors have shown that the boronate derivatives also react with oxidants (H2O2, NaOCl, and ONOO−) that are generated in tumor mitochondria and the tumor microenvironment by HPLC, but have not taken any further investigations to study the consequence of this reaction in cancer cells, thus no results to support any claim in discussion e.g. these novel polyphenolic derivatives may enhance the scope of cancer immunotherapies’.

Response: As mentioned above, these polyphenolic boronates are unique in that they are able to form the parent compounds (honokiol, Mito-honokiol, magnolol, or Mito-magnolol) following their reaction with oxidants. However, in order to show that reactive oxygen and nitrogen species are involved, the polyphenolic boronates should be tested in appropriate in vivo cancer xenografts using the pancreatic or melanoma cancer cells. We took the first step in this study showing that mitochondria-targeted polyphenolic boronates inhibit tumor cell proliferation.   

3. The main results of this manuscript focus on the chemical reactions that form borate derivatives, which are not really cancer-related findings.

Response: We disagree with the reviewer that this manuscript only focuses on chemical reactions. Increasing evidence suggests that the oxidative and nitrative tumor microenvironment decreases the efficacy of immunotherapy, causing cytokine nitration, T cell exhaustion, and so forth (Calì B et al. Front Immunol. 2021 Oct 5;12:718098 and De Sanctis F et al. J Immunother Cancer. 2022 Jan;10(1):e003549). As shown in many of our publications, boronates are nearly a million times faster in reacting with peroxynitrite than hydrogen peroxide. Boronates are able to directly scavenge oxidants. The involvement of reactive nitrogen species was shown in KPC mice. It is our hope that the redox-based mitochondria-targeted polyphenolic boronates will be tested in the appropriate mice models, so that they may advance our understanding of immunosuppressive oxidants in immunotherapy and immuno-oncology.   

Round 2

Reviewer 2 Report

Thank you for responding to all of our reviews, Good Job67 / 5.000

Risultati della traduzione

R

Author Response

We thank Reviewer 2 for the compliment.

Reviewer 3 Report

About response 1. Author mentioned that Thus, we have used totally new redox-active boronate derivatives of both mitochondria-targeted and untargeted polyphenols´ in response 1. If it's a completely new derivative, is there any evidence that this new derivative still targets the mitochondria? 

Regarding Response 2, the authors mention "Oxidant formation in the tumor microenvironment" several times in the title, abstract and text, but in this manuscript the authors only do it in a cell-free system that is very different from the tumor microenvironment reactions, leading to the question whether the new derivatives also mitigate the formation of oxidants in the tumor microenvironment. If the authors want to claim that it is the tumor microenvironment, they should compare the oxidant levels in cells vehicle VS treated using HPLC.

Regarding Response 3, I agree with it. 

Author Response

About response 1. Author mentioned that ’Thus, we have used totally new redox-active boronate derivatives of both mitochondria-targeted and untargeted polyphenols´ in response 1. If it's a completely new derivative, is there any evidence that this new derivative still targets the mitochondria?

Response: We showed in the Results section that the IC50 value of mitochondria-targeted polyphenolic boronates is nearly similar to the that of mito-polyphenol. Based on our extensive experience with mitochondria-targeted drugs and mitochondrial respiration studies using Seahorse technology, we are confident that mito-polyphenolic boronates target mitochondria and inhibit mitochondrial respiration and elicit a similar type of toxicity in cancer cells.

Regarding Response 2, the authors mention "Oxidant formation in the tumor microenvironment" several times in the title, abstract and text, but in this manuscript the authors only do it in a cell-free system that is very different from the tumor microenvironment reactions, leading to the question whether the new derivatives also mitigate the formation of oxidants in the tumor microenvironment. If the authors want to claim that it is the tumor microenvironment, they should compare the oxidant levels in cells vehicle VS treated using HPLC.

Response: As indicated in the previous response, in order to realistically test the effect of polyphenolic boronates in the tumor microenvironment, it is necessary to use an appropriate mice model (PPC) and investigate the effect of mitochondria-targeted boronates on nitration of cytokines. We don’t have the experience or expertise to measure nitrated cytokines. We have included a sentence in the conclusion that future studies should address this aspect.